# Harnessing machine learning to guide phylogenetic-tree search algorithms

Dana Azouri [1,2], Shiran Abadi [1], Yishay Mansour[3], Itay Mayrose [1✉] & Tal Pupko [2✉]

Inferring a phylogenetic tree is a fundamental challenge in evolutionary studies. Current paradigms for phylogenetic tree reconstruction rely on performing costly likelihood optimizations. With the aim of making tree inference feasible for problems involving more than a handful of sequences, inference under the maximum-likelihood paradigm integrates heuristic approaches to evaluate only a subset of all potential trees. Consequently, existing methods suffer from the known tradeoff between accuracy and running time. In this proof-of-concept study, we train a machine-learning algorithm over an extensive cohort of empirical data to predict the neighboring trees that increase the likelihood, without actually computing their likelihood. This provides means to safely discard a large set of the search space, thus potentially accelerating heuristic tree searches without losing accuracy. Our analyses suggest that machine learning can guide tree-search methodologies towards the most promising candidate trees.

[1] School of Plant Sciences and Food Security, Tel Aviv University, Ramat Aviv, Tel-Aviv, Israel. [2] The Shmunis School of Biomedicine and Cancer Research, Tel Aviv University, Ramat Aviv, Tel-Aviv, Israel. [3] Balvatnik School of Computer Science, Tel Aviv University, Ramat Aviv, Tel-Aviv, Israel. ✉email: itaymay@tauex.tau.ac.il; talp@tauex.tau.ac.il

One of the most fundamental goals in biology is to reconstruct the evolutionary history of all organisms on earth. The obtained phylogeny is of interest to many downstream analyses concerning evolutionary and genomics research. Until recently, most studies focused on a few to several dozens of sequences. Current phylogenomic studies analyze longer sequences (up to entire genomes) and include a greater diversity (hundreds and even thousands of lineages), consequently challenging the ability of computational resources to handle these amounts of data.

Leading approaches for phylogeny reconstruction rely on probabilistic evolutionary models that describe the stochastic processes of nucleotide, amino-acid, and codon substitutions[1]. Given an evolutionary model, a tree topology with its associated branch lengths, and a multiple sequence alignment, the likelihood of the data is efficiently computed using Felsenstein's pruning algorithm[2]. While the alignment is usually assumed to be known, parameters of the evolutionary model, the tree topology, and its associated branch lengths are often inferred by maximizing the likelihood function. Thus, for a specific evolutionary model with fixed parameter values, each tree inference algorithm visits a large number of candidate tree topologies and for each such topology, it searches for the optimal set of branch lengths. Notably, the number of possible tree topologies increases super-exponentially with the number of sequences. Moreover, the computational search for the maximum-likelihood tree topology was shown to be NP-hard[3]. Optimizing the set of branch lengths for each candidate tree is computationally intensive, adding another layer of complexity to this endeavor. Thus, all current algorithms for phylogenetic tree reconstruction use various heuristics to make tree inference feasible.

The general approach for a maximum-likelihood heuristic search is to begin either with a random starting tree or with a starting tree obtained by rapid and generally less accurate methods such as Neighbor Joining[4,5]. The score of this initial tree is its log-likelihood, which is based on the specified probabilistic model. Next, a set of alternative topologies is considered, each of which is a small modification of the current tree topology (each such topology is considered to be a "neighbor" of the current topology). The neighbor with the highest score is selected and used as an initial tree for the next step. The process proceeds iteratively until none of the alternative trees produces a higher score compared to the current one. Various algorithms differ in their definition of a neighbor. In this study, we focus on subtree pruning and regrafting (SPR)[6]. An SPR neighbor is obtained by pruning a subtree from the main tree and regrafting it to the remaining tree, as illustrated in Fig. 1. Several improvements to the basic heuristic scheme described above have been suggested. These improvements include better exploration of the tree space and the introduction of shortcuts in order to substantially reduce running time with little to no influence on inference accuracy. Notable examples include: (1) proceeding with the first neighbor that improves the likelihood score without examining the remaining neighbors[7]; (2) avoiding optimization of the entire branch lengths by optimizing only those in the vicinity of the regrafted subtree[7]; (3) discarding neighbors whose estimated sum of branch lengths highly deviates from that of the current tree[8]; (4) genetic algorithms and simulated annealing versions of the heuristic search[9,10]. In addition, a common practice is to apply the bootstrap procedure that provides a measure of confidence for each split in the obtained tree. This is done by executing the tree search on bootstrapped data at least 100 times. This time-consuming step further emphasizes the need for efficient heuristics[3,11]. To our knowledge, machine-learning tools have not been employed for enhancing the heuristic tree search.

In this study, we use a diverse set of thousands of empirical datasets to train a supervised machine-learning regression model, specifically a random forest learning algorithm, in order to predict the optimal move for a single step in a phylogenetic tree search. The output of this learner, trained on a succinct collection of 19 features, is a numerical value for each possible SPR move that represents its propensity to be the highest-scoring neighbor. Our results show that this procedure yields very high agreement between the true and inferred rankings, indicating the high predictive power of the developed machine-learning framework. Furthermore, we demonstrate that using the learning framework it is sufficient to evaluate the costly likelihood score for a small subset of all possible neighbors. This study thus establishes a comprehensive proof-of-concept that methodologies based on artificial intelligence can substantially accelerate tree-search algorithms without sacrificing accuracy.

## Results

**A machine-learning algorithm for accelerating the maximum-likelihood tree search.** Our goal was to rank all possible SPR neighbors of a given tree according to their log-likelihood without actually computing the likelihood function. To this end, we relied on a set of features that can be efficiently computed and thus capture essential information regarding the tree and the proposed SPR rearrangements. Specifically, we trained a machine-learning algorihm, random forest regression, to predict the ranking of all possible SPR modifications according to their effect on the log-likelihood score. The algorithm was trained on a large set of known examples (data points). In our case, each data point is a pair $(\mathbf{V}, L)$. $\mathbf{V}$ is an array that includes the starting tree, the resulting tree following an SPR move, and the set of features, while $L$ is a function of the log-likelihood difference between the starting and the resulting tree (see "Methods" section). The regression model learns the association between $\mathbf{V}$ and $L$. Given a trained algorithm and a starting tree topology, an array $\mathbf{V}$ is computed for each possible SPR move. The trained machine-learning algorithm provides the ranking of all possible SPR moves according to their predicted $L$ values. A perfect machine-learning model would predict the optimal SPR neighbor and would thus eliminate the need for expensive likelihood computations. A sub-optimal predictor may also be highly valuable if the vast majority of the SPR moves can be safely discarded without computing their likelihoods.

The machine-learning algorithm was trained on 20,880,8151 data points, one data point for each possible SPR move of 4200 different empirical phylogenies. The empirical alignments varied in terms of their attributes, e.g., the number of sequences (7 to 70), the number of positions (62 to 10,000), and the extent of sequence divergence (Supplementary Fig. 1). The number of neighbors of each tree is affected by the number of sequences and by the tree topology and ranges between a few dozens to over ten thousand. We chose to analyze empirical rather than simulated data, as it is known that reconstructing the best tree is more challenging for the former[12–14]. The learning was based on 19 features, extracted from each data point (Table 1). Some features were extracted from the starting trees, e.g., the lengths of the branches in the pruning and regrafting locations, while others were generated based on the subtrees induced by the SPR move, e.g., the sum of branch lengths of the pruned subtree (Fig. 1).

**Performance evaluation.** We evaluated the performance of our trained learner in a ten-fold cross-validation procedure. Namely, the empirical datasets were divided into ten subsets, such that in each of the ten training iterations, the induced data points of nine folds were used for training the model, and the remaining data

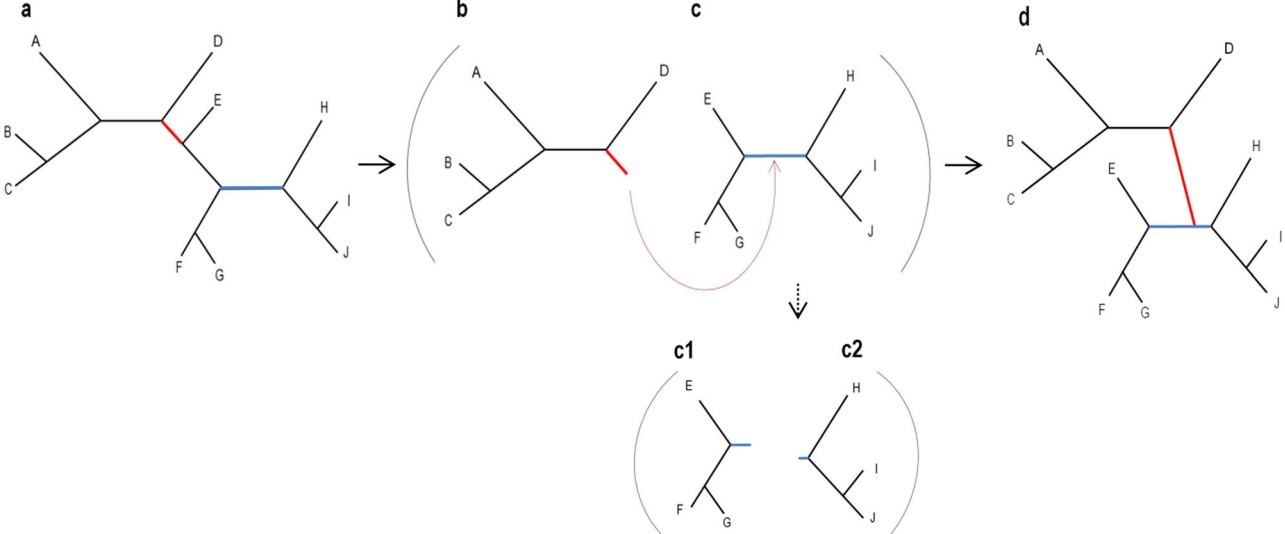

**Fig. 1 The trees defined by an SPR move.** For each data sample, two trees and four subtrees were considered: **a** an example of a starting tree; **b**, **c** the two subtrees induced by the pruned branch of a tree *a* (in red), where *b* is the pruned subtree and *c* the remaining subtree; **c1**, **c2** the regrafted branch (in blue) also induces two subtrees, from both sides of the regrafted branch of subtree *c*; **d** the resulting tree following the SPR move.

| Table 1 Features used in the machine-learning framework. | | | | |
|---|---|---|---|---|
| **# Feature** | **Feature name** | **Details** | **Represented action** | **Tree considered** |
| 1 | Total branch lengths | The sum of branch lengths in the starting tree | Shared for pruning and regrafting | Initial tree (a in Fig. 1) |
| 2 | Longest branch | The length of the longest branch in the starting tree | | |
| 3–4 | Branch length | The length of the branch that was being pruned or regrafted | Both pruning and regrafting | |
| 5 | Topology distance from the pruned node | The number of branches in the path between the regrafting and the pruning branches, not including these branches | Regrafting only | |
| 6 | Branch length distance from the pruned node | The sum of branches in the path between the regrafting and the pruning branches, not including these branches | | |
| 7 | New branch length | The approximated length of the new branch formed due to pruning (see Supplementary Note 1 for feature extraction details) | | |
| 8–11 | Number of species | The number of leaves in the four subtrees | Both pruning and regrafting | Each of the four subtrees (b, c, $c_1$, $c_2$ in Fig. 1) |
| 12–15 | Total branch lengths | The sum of branch lengths in the four subtrees | | |
| 16–19 | Longest branch | The length of the longest branch in the four subtrees | | |

The table lists the 19 features on which the machine-learning algorithm is based, extracted for each data point. Features 1-7 are extracted from the starting tree, while the remaining features are extracted from the four subtrees in Fig. 1. Features 1 and 2 are not affected by SPR moves.

points were used for testing. We first evaluated the accuracy of the model in ranking alternative SPR moves. The Spearman rank correlation coefficient ($\rho$) was thus computed between the true ranking, inferred through a full likelihood-optimization, and the predicted ranking, based on the machine-learning predictions. The mean $\rho$, averaged over all 4200 samples, was 0.91 (Fig. $2a_1$), suggesting that the machine-learning algorithm successfully discriminates between beneficial and unfavorable SPR moves.

Notably, the Spearman correlation quantifies the prediction performance when all SPR neighbors are considered. However, in a typical hill-climbing heuristic, the single best SPR neighbor is chosen as the starting tree for the next step. It is thus interesting to estimate the ability of the algorithm to predict this best neighbor. Accordingly, we measured the performance of the trained algorithm by two additional metrics: (1) the rank of this best move within the predicted ranking; (2) the rank of the predicted best move within the true ranking, as obtained according to the full likelihood optimization. In 81% and 95%

of the datasets, the best move was among the top 10% and 25% predictions, respectively (Fig. $2a_2$). In 95% and 99% of the datasets, the top-ranked prediction was among the top 10% and 25% SPR moves, respectively (Fig. $2a_3$). Moreover, in 99.99% of the cases, the top prediction resulted in a higher likelihood compared to the starting tree, suggesting that an improvement is typically obtained. In contrast, a random move increased the likelihood score in only 2.1% of the datasets. These results suggest that the machine-learning algorithm can direct the tree search to a narrow region of the tree space, thus avoiding numerous expensive likelihood calculations.

We next evaluated the trained model on entirely different datasets than the training data (see "Methods" section). Unlike the training data, the machine-learning algorithm was not optimized on these data, not even in cross-validation, thus negating possible overfitting effects. When applied to these validation data, the performance of the trained model was very similar to that reported above using cross-validation (average $\rho = 0.9$; Fig. $2b_{1-3}$),

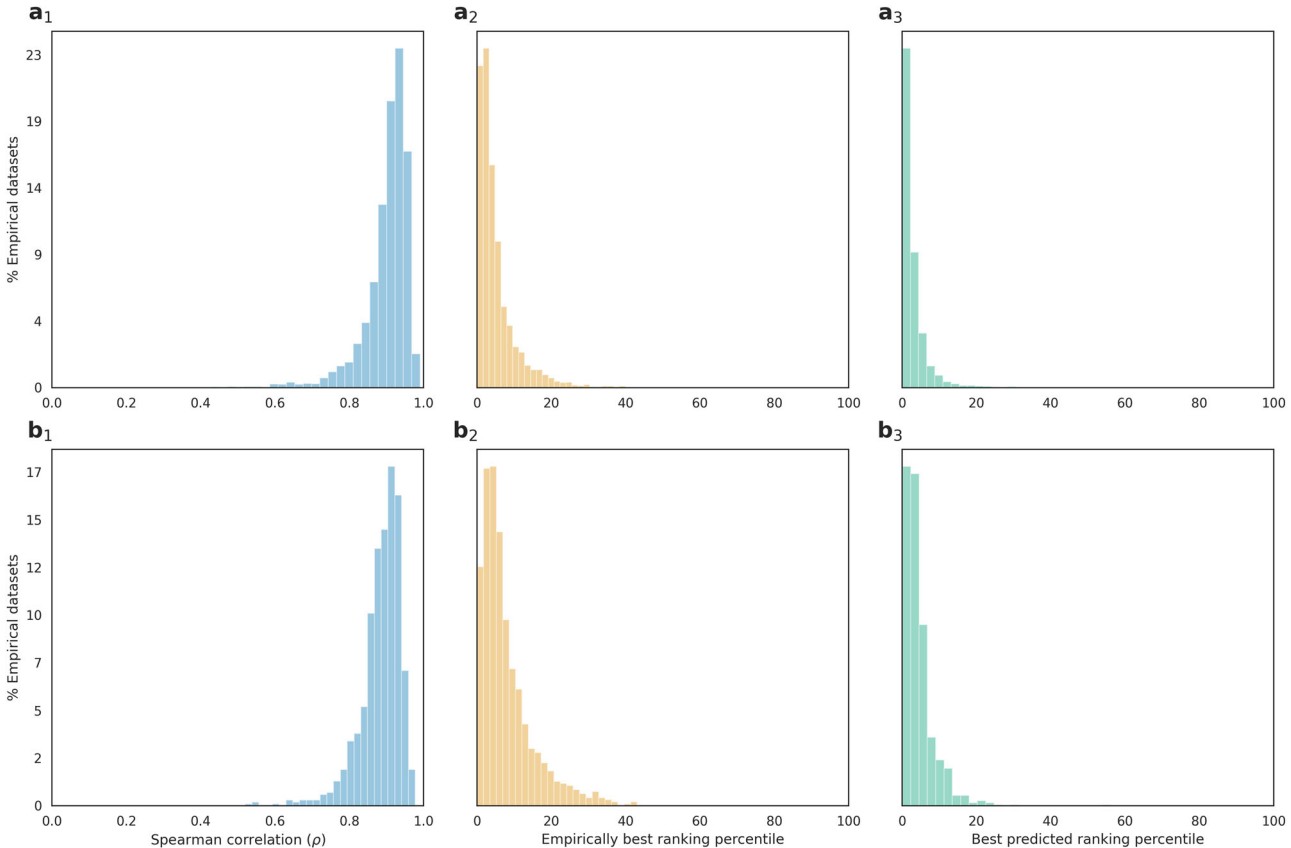

**Fig. 2 Performance evaluation scores on empirical datasets.** A histogram of the three performance scores of the learning algorithm evaluated on: **a** the 4200 starting trees, using cross-validation; **b** the 1000 validation set starting trees. On the Y axis of both panels a and b are the percentages of empirical datasets in each accuracy score bin: (1) accuracy is computed as the Spearman correlation coefficient between the predicted ranking and the true ranking of neighboring trees; (2) accuracy is the rank (in percentile) of the empirically best neighbor within the predicted ranking; (3) accuracy is the rank (in percentile) of the predicted best neighbor within the empirical ranking.

suggesting that the machine-learning algorithm is well generalized for various datasets, evolved under an array of evolutionary scenarios.

To gain further insight into factors affecting the prediction accuracy, we compared the accuracy across the six databases used (four for training and two for validation). Among the six databases, predictions were most accurate for Selectome, with a mean $\rho$ of 0.95, and least accurate for ProtDBs, with a mean $\rho$ of 0.83 (Supplementary Fig. 1a). In addition, we analyzed whether the prediction accuracy is affected by: (1) the number of taxa; (2) the level of divergence as measured by the sum of branch lengths; (3) the alignment length; (4) the percentage of gap characters in the alignment; (5) the deviation from ultrametricity as measured by the MAD score, which quantifies departures from ultrametricity[15]. The most meaningful correlation ($r^2 = 0.23$) was observed between $\rho$ and the level of divergence: for trees with more than 49 sequences in the validation set, the predictions tended to be less accurate for highly diverged trees (Supplementary Fig. 1b–f). Finally, we tested whether increasing the number of alignments analyzed within the training data could further increase the prediction accuracy. Increasing the number of trained samples from 4200 to 6000 did not significantly increase the accuracy ($P$-value > 0.97 using one-way ANOVA; Supplementary Fig. 2).

**The effect of learning using an oversimplified model.** We repeated the above learning and testing procedure with the

Jukes and Cantor (JC) model, which assumes that all types of substitutions are equally likely and ignores site rate variation. Thus, this model is substantially simpler than the GTR + I + G model used in the original model. When both learning and testing were performed assuming the JC model, the accuracy of the machine-learning model was high (average $\rho = 0.89$), similar to the accuracy obtained for the GTR + I + G model (average $\rho = 0.91$). We also evaluated the performance when the training was performed under the JC model, and the test data comprised of log-likelihoods computed under the GTR + I + G model. Under these conditions, the accuracy was only slightly lower (average $\rho = 0.88$; Supplementary Fig. 3a). The results obtained when alternative accuracy metrics were considered are detailed in Supplementary Table 1 and Supplementary Fig. 3b, c. These results suggest that learning on an oversimplified substitution model is not detrimental for discriminating among potential neighboring trees, even when the underlying model is more complex than that used for training.

**Performance evaluation on an example dataset: protein-coding genes in algae.** We exemplify the application of the machine-learning algorithm on a specific dataset, consisting of 28 algae protein-coding genes (see "Methods" section). We reconstructed a neighbor-joining starting tree, generated all its 2462 SPR neighbors, and ranked them according to their log-likelihoods. We then compared this ranking to the ranking predicted by the trained machine-learning algorithm. The Spearman rank

correlation ($\rho$) between the true and the predicted rankings was 0.93, which is similar to the average $\rho$ reported for both the training and validation data. Indeed, the best move was among the top four predictions, and the top SPR move predicted by the model was the sixth-best possible move. Furthermore, the best SPR move and the predicted best SPR move chose to prune the same clade of the phylogenetic tree (i.e., they only differ in the regrafting position).

While the ultimate goal is to predict the ranking of the possible SPR moves in order to limit the search space, focusing on one example enables the inspection of the actual predicted change in log-likelihood between each potential resulting tree and the starting tree. For this example, a Pearson correlation ($r^2$) of 0.94 between the predicted and true change in log-likelihood was observed (the full list of the predicted and true log-likelihood differences for all 2462 single-step SPR moves is given in Supplementary Data 1). The predicted best move improved the initial tree by 25.6 log-likelihood points, whereas the improvement obtained by the best SPR move was 31.23 log-likelihood points. Moreover, according to our model, 19 and 2443 SPR moves were predicted to increase and decrease the log-likelihood, respectively, and these predictions were true for 95% and 98% of these cases. These results corroborate the potential of the machine-learning approach to correctly discard many irrelevant SPR neighbors.

In addition, we measured the running time for evaluating the 2462 neighboring trees for this example. The computation of the features and the application of the trained model for each neighbor took $2 \times 10^{-4}$ s on average. The likelihood computation (with branch lengths optimization) took 0.15 s on average for each neighbor, roughly 750 times longer compared to the machine-learning algorithm.

We next examined whether the high performance of the trained model is maintained when applied to other intermediate trees in the chain towards the maximum-likelihood tree. When applied to the second phase of the search, i.e., starting from the best possible neighbor of the initial tree, the trained model yielded results that are highly similar to those reported for the initial tree (Spearman correlation coefficient of $\rho = 0.9$). The best move according to the predictions increased the true log-likelihood score by 25.9, implying that the likelihood improvement is maintained following additional SPR steps. Finally, we examined the algorithm performance when the initial tree is one step away from the maximum-likelihood tree. To this end, we applied the machine-learning algorithm for each of the 2492 SPR neighbors of the maximum-likelihood tree. The model predicted the maximum-likelihood tree to be among the top five predictions in 98% of the cases.

We next studied the applicability of the machine-learning algorithm within a straightforward tree-search heuristic. Starting from the neighbor-joining tree, we evaluated the likelihood of all top predicted 5% SPR moves and then moved to the highest scoring tree. We repeated this process until no improvement in log-likelihood was obtained. The log-likelihood increased and the Robinson-Foulds (RF)[16] distance monotonically decreased for 15 consecutive moves (Fig. 3). This procedure probably recovered the global maximum-likelihood tree (the tree with the highest likelihood obtained when running PhyML, RaxML-NG, and our own implementation from multiple starting points).

**Performance evaluation on more complex datasets.** We further validated that the accuracy of our model remains high when applied to datasets that represent larger and more complex biological scenarios than the ones included within the data used to train and test our model. To this end, we analyzed a partitioned

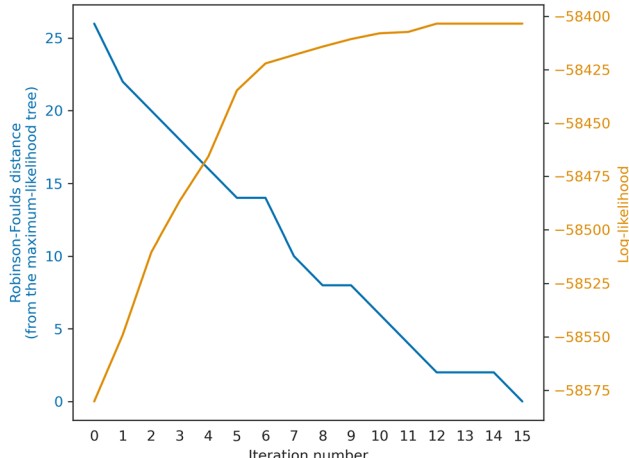

**Fig. 3 Example of an iterative chain of moves.** Evaluation metrics for the convergence behavior of an iterative chain of trees towards the maximum-likelihood tree. The chain was initiated with the Neighbor-Joining tree reconstructed for the protein-coding algae dataset we used as an example throughout the manuscript (iteration #0). The line plots represent the Robinson-Foulds distance from the maximum-likelihood tree (in blue; left $Y$ axis), and the log-likelihood of the tree obtained in each iteration (in orange; right $Y$ axis).

dataset, consisting of eight protein-coding genes belonging to 59 plant species, where each partition is characterized by a different set of GTR + I + G model parameters, and the branch lengths of each partition are based on the proportional model[17]. In this case, the best move was among the top three predictions (i.e., within the top 0.025% predictions) and the best-predicted move was the twelfth possible move (i.e., top 0.1% SPR moves); the overall correlation between the predicted and true rankings was 0.74.

We next evaluated the performance of the machine learning approach on a dataset with a much larger number of species (403) than those used for training (spanning 7–70 species). For this dataset, the starting neighbor-joining tree has 624,508 SPR neighbors. In this case, the best move was among the seven top predictions, which falls within the top 0.001% predictions, and the best prediction was among the top 20 moves (within the top 0.003% possible SPR neighboring-trees); the overall correlation between the predicted and true rankings was 0.69.

**Feature importance.** Feature importance analysis quantifies the relative contribution of each feature to the prediction accuracy. In our implementation, the feature that contributed most to the prediction accuracy was the sum of branch lengths along the path between the pruning and the regrafting locations, while the second-best feature was the number of nodes along that path. These findings provide some justification for the common practice of considering only local changes in various tree search heuristics[7,9,18]. The next three features were the sum of branch lengths of the starting tree, the length of the pruned branch, and the length of the longest branch in the pruned subtree (for the important values of all features, see Supplementary Table 2).

Many common tree-search heuristics utilize a single feature to limit the scope of inspected neighbors. We thus exploited the devised framework to examine whether the use of a single feature leads to similar performance. To this end, we trained 19 random forest models on the training set, such that each model accounted for a single feature. The performance of each of these models provided a measure of the predictive power of each feature, independent of the others. The best single-feature model obtained

a Spearman correlation coefficient of $\rho = 0.69$ on average across the training set and was based on the number of nodes in the path between the pruning and the regrafting locations, a feature that was ranked second when the entire set of features was used for training. The average $\rho$ across the training set for all the other features was below 0.28 (Supplementary Table 3). These observations, together with the substantial increase in average $\rho$ when comparing the usage of a single feature to using the entire set of features combined (average $\rho$ of 0.91), highlights the benefit of relying on a large set of features that together provide a more informative prediction.

## Discussion

Inferring a phylogenetic tree is of central importance in numerous evolutionary studies. As follows, methods for tree reconstruction are widely used by the biological research community. Still, since such methods incur complex computations, all existing methods attempt to reduce running time at the expense of accuracy, being dependent on heuristics to overcome the feasibility problem. Here we developed a machine-learning framework, trained to rank neighboring trees according to their propensity to increase the likelihood. The evident high predictive power of this framework demonstrates that the computationally-intensive step of likelihood evaluation can be limited to a small set of potential neighbors, substantially reducing the running time without jeopardizing accuracy. By boosting tree inference, our study directly impacts efforts of downstream analyses, such as molecular dating[19], inference of positive selection[20], protein fold recognition[21], identification of functionally divergent protein residue[22], recombination detection[23], and ancestral sequence reconstruction[24]. Furthermore, our research could grant the development of richer and more realistic substitution models, which are currently too computationally intensive to be considered within a tree-search procedure (e.g., a covarion model[25] for codon characters). This hypothesis is based on the partitioned dataset analyzed in our study, and on our experiment in which high performance was still observed when we applied a machine-learning model trained under the JC model[26] to data evaluated under the GTR + I + G model.

Ranking of neighboring trees to speed up the tree search was previously suggested, albeit with the use of a single attribute and without learning from large training data. For example, Hordijk and Gascuel[8] proposed testing only neighbors for which their estimated total sum of branch lengths does not substantially differ from the starting tree. Our methodology advances over previous approaches, as we use multiple features instead of one, and utilize machine learning to optimally combine these features based on extensive training. Notably, a recent study suggested the use of deep neural networks to classify alignments as being either Felsenstein-type or Farris-type[27]. Moreover, Suvorov et al.[28] and Zou et al.[29] utilized convolutional and residual neural networks, respectively, to infer unrooted four-taxa topologies from multiple sequence alignments. While their devised methods perform well, they can currently be applied to infer topologies of four taxa only. In addition, in order to reconstruct the true generating topology, they were required to rely on simulated datasets, which were previously shown to be easier to interpret and infer[12–14]. The objective of our study, narrowing the search space in a single step towards a final, faster, convergence of the maximum likelihood, enabled us to rely on empirical datasets for training and testing.

How can our machine-learning algorithm be used in practice? One trivial application would be to start evaluating the neighboring trees, starting from the top-ranked predicted neighbor. If this neighbor obtains a log-likelihood score that is higher than the starting tree, proceed with that tree as the starting tree, iteratively repeating this procedure. If this neighbor obtains a log-likelihood score that is lower than the starting tree, evaluate the next ranked neighbor. End the iterative chain of tree search when no improvement is obtained. A similar procedure could be applied by evaluating the log-likelihoods of the set of 5% top predictions and progressing with the best among it. Clearly, more sophisticated tree search schemes can be considered. For example, one could progress a few steps, based on the best predictions only, without evaluating the likelihoods, expecting the obtained tree to have a higher log-likelihood compared to the starting tree. Furthermore, our approach can be integrated within existing maximum-likelihood frameworks, which are already implemented in the leading tree search algorithms, such as, RAxML[30], PhyML[31], and IQtree[32]. For example, in IQtree a set of trees is kept and the algorithm samples from this set. Such an approach to sampling within a subset of more likely neighbors can easily be combined with our machine-learning approach that allows sampling the most promising trees while rapidly traversing large regions of the tree space. Further developments of the proposed methodology towards a complete search are possible. For example, we have not put the effort into assessing the branch lengths associated with the inferred topology or in predicting log-likelihoods of trees under different parameters-optimization schemes. It is also interesting to further study how our approach generalizes to additional substitution models of evolution, such as amino-acid models codon models, and additional partition models[33,34]. Furthermore, the convergence behavior in regions of the tree space with high likelihood requires more robust investigation than the anecdotal evidence we provided in this study. In addition, our algorithm was implemented using SPR moves only. The benefit of using additional types of tree rearrangement moves, such as nearest-neighbor interchange (NNI)[35,36] and tree bisection and regrafting (TBR)[37] should be evaluated.

To conclude, we provide a methodology that can substantially accelerate tree-search algorithms without sacrificing accuracy. We believe that harnessing artificial intelligence to the task of phylogenomics inference has the potential to substantially increase the scale of the analyzed datasets and, potentially, the level of sophistication of the underlying evolutionary models.

## Methods

**Empirical and validation data.** We assembled training data composed of 4200 empirical alignments from several databases: 3894 from TreeBase[38], 151 from Selectome[39], 45 from protDB[40], and 110 from PloiDB[41]. TreeBase is a repository of user-submitted phylogenies; Selectome includes codon alignments of species within four groups (Euteleostomi, Primates, Glires, and Drosophila); protDB includes genomic sequences that were aligned according to the tertiary structure alignments of the encoded proteins published in BALIBASE[42]; and PloiDB contains alignments with sequences belonging to a single plant genus and a potential outgroup. We randomly selected datasets with 7 to 70 sequences and more than 50 sites, excluding alignments containing sequences that are entirely composed of gapped or missing characters.

To test the predictive power of our model also over unseen validation data that were neither used for training our model nor for cross-validation, we gathered a database encompassing 1000 multiple sequence alignments, collected from two databases that were not used to generate the training set: 500 datasets from PANDIT[43], which includes alignments of coding sequences, and 500 datasets from OrthoMaM[44], a database of orthologous mammalian markers. Next, we verified that our validation set is composed of a variety of biological data attributes (Supplementary Fig. 1).

*Example datasets.* The example dataset that we used to exemplify the main results of our study was composed of 28 *Algae* protein-coding plastid sequences, composed of four genes (*psa*A, *psa*B, *psb*C, and *rbc*L), as obtained in Lewis et al.[45]. Next, we used an additional example dataset as one that reflects a more complex model. This empirical multi-gene alignment was composed of eight partitions of 59 plant sequences (one partition for each gene), as obtained in Kobert et al.[46]. The partitioned model assigned a distinct GTR + I + G substitution model for each partition, assuming the proportional (namely "scaled") branch linkage model. Finally, we used a dataset with 403 species as additional validation for our machine-learning model robustness in terms of the number of species in the

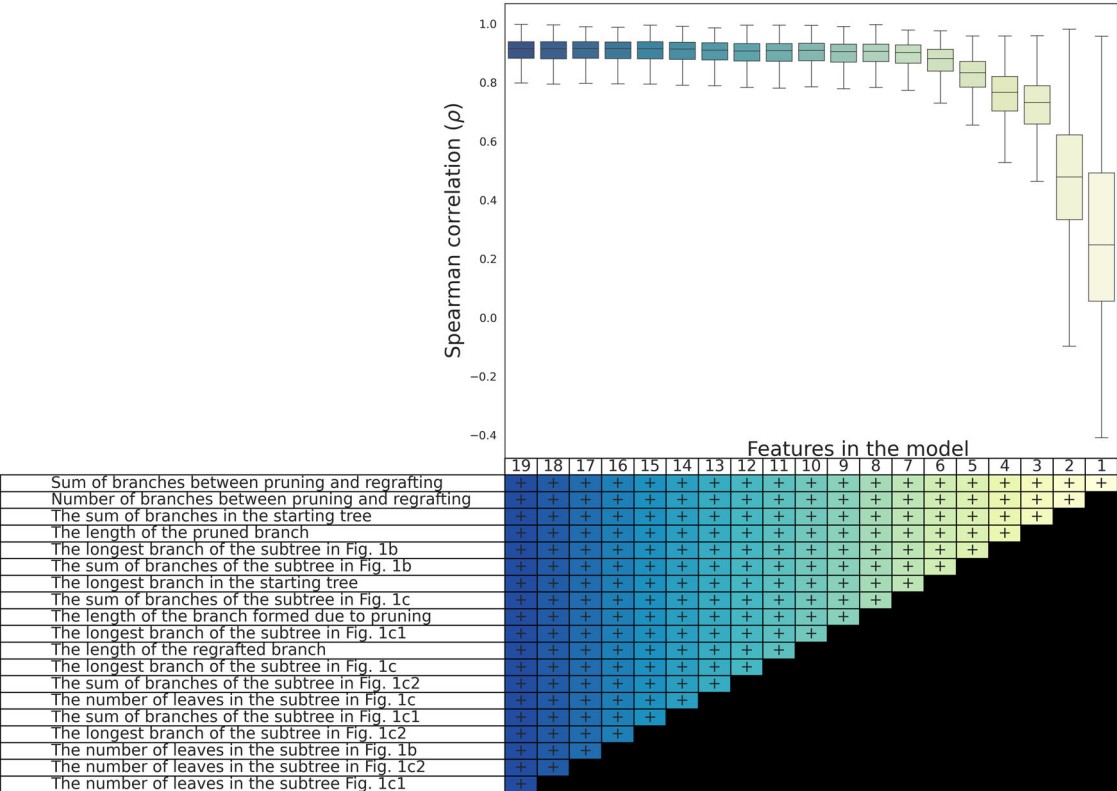

**Fig. 4 Feature selection following a backward stepwise elimination procedure.** The mean Spearman correlation coefficient obtained when using a decreasing number of features for training the algorithm on the 4200 starting trees. The box shows the quartiles of the dataset (thus the center is the median) while the whiskers extend to show the 1.5 × IQR past the low and high quartiles. The table at the bottom elaborates the feature composition within each set of features, as determined by the backward stepwise elimination procedure.

alignments used for training the model. This alignment was obtained from PANDIT[43].

*Starting trees reconstruction, SPR neighbors generation, and likelihood estimation.* The starting tree for each alignment was reconstructed using BioNJ[47] as implemented in PhyML 3.0[31], assuming the GTR + I + G model. We optimized the branch lengths for each starting tree and all its SPR neighbors using RAxML-NG[48]. The substitution rate parameters were optimized for the starting tree and were fixed for all neighbors, i.e., we recorded the log-likelihoods of the neighboring trees assuming the GTR + I + G optimized parameters of the starting tree.

**A machine-learning algorithm for ranking neighboring trees.** Random forest for regression, as implemented in Python Scikit-learn module[49], was applied using 70 decision trees. In each split of the tree, a random subset of one-third of the total number of features was considered. The target value of the machine-learning training was computed as $target = \frac{LL_{neighbor} - LL_{starting tree}}{LL_{starting tree}}$, namely, the log-likelihood difference between the neighbor and its starting tree, divided by the log-likelihood of the starting tree. Notably, these ratios are log distributed across the training set and may lead to unbalanced decision trees in the random-forest training. Therefore, the training outcomes were transformed according to $f(target) = 2^{target+1}$ to generate a distribution that is more uniform (Supplementary Fig. 4). The reversed transformation was applied to the predicted values accordingly.

The learning scheme we implemented in this study is a random forest regression algorithm. This model was chosen over four other alternative supervised-machine-learning regression algorithms we implemented, as it outperformed all others: Support vector machine, Bayesian Ridge, Lasso, and K-Nearest-Neighbors (Supplementary Table 4).

**Predictive features.** The learning was based on extracting 19 features from each data point (Table 1). The computation of all features was implemented in Python and required O($n\log n$) operations for all the pruning and regrafting locations of a single tree, $n$ being the number of sequences (see Supplementary Note 1 for feature extraction details). The first seven features were extracted from the starting trees (Fig. 1, Table 1; features 1–7). The remaining features rely on the following definition of four intermediate subtrees: the two subtrees induced by splitting the starting tree at

the pruning location and the two subtrees induced by splitting the remaining subtree at the regrafting location (Fig. 1). For each of these four subtrees, we calculated three features, resulting in a total of twelve features (Table 1; features 8–19).

To examine whether the feature set could be reduced to enhance computational performance, we applied a backward stepwise elimination procedure[50]. To this end, we began with the full set of 19 features. We then removed the feature with the minimal importance score and trained the random forest algorithm for the remaining features, to compute the $\rho$ metric. We repeated this procedure, successively eliminating an additional feature with the minimal importance score (Fig. 4). The best $\rho$ value was obtained when all the features were included. Only when using 14 or fewer features, a statistically significant reduction in accuracy was detected (P-value < 0.02 and P-value > 0.49, for one-sided t-test for the means when using 14 and 15 features to 19, respectively). The results across the entire analyses are presented using the entire set of features.

**Reporting summary**. Further information on research design is available in the Nature Research Reporting Summary linked to this article.

## Data availability
The datasets contained within the empirical set have been deposited in Open Source Framework (OSF) with the identifier DOI 10.17605/OSF.IO/B8AQJ[51]. These datasets were assembled from the following databases: TreeBase (https://treebase.org/treebase-web/urlAPI.html); Selectome (https://selectome.org/); protDB (https://protdb.org/); PloiDB (https://doi.org/10.3732/ajb.1500424); PANDIT (https://www.ebi.ac.uk/research/goldman/software/pandit); OrthoMaM (https://orthomam.mbb.cnrs.fr/).

## Code availability
The code that supports the findings of this study was written in Python version 3.6 and has been deposited in Open Source Framework (OSF) with the identifier DOI 10.17605/OSF.IO/B8AQJ[51]. Computation of likelihoods and parameter estimates were executed using the following application versions: PhyML 3.0[31], RAxML-NG 0.9.0[48].

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

## Acknowledgements

We acknowledge the Data Science & AI Center at TAU for supporting this study. D.A. was supported by The Council for Higher Education program for excellent Ph.D. students in Data Sciences and by a fellowship from the Fast and Direct Ph.D. Program at Tel Aviv University. S.A. was supported by the Rothchild Caesarea Foundation and by a fellowship from the Edmond J. Safra Center for Bioinformatics at Tel Aviv University. Y. M. was supported in part by a grant of the Israel Science Foundation (ISF) 993/17. I.M. was supported by an Israel Science Foundation grant 961/17. T.P. was supported by an Israel Science Foundation grant 802/16.

## Author contributions

D.A., S.A., Y.M., I.M., and T.P. designed the study, helped in interpreting the results, and provided inputs on the draft. D.A. implemented the pipeline, performed the analyses, and drafted the manuscript. Y.M., I.M., T.P. supervised this work and revised the manuscript.

## Competing interests

The authors declare no competing interests.
