## [Peer Review File · Nature Communications]

Reviewers' Comments:

Reviewer #1:

Remarks to the Author:

The authors present a machine learning framework to predict the log likelihood improvement of SPR moves in phylogenetic tree searches with the goal to avoid compute-intensive likelihood calculations by identifying promising candidate SPR moves.

Overall, the paper is very well written and technically sound.

Having worked on phylogenetic tree search algorithms for over 15 years, I have little to complain about, since the vast majority of questions I had in mind or omissions I thought were made, were actually answered in the following paragraph.

This approach might have a good potential for accelerating likelihood-based tree searches, but needs to be implemented and tested in one or several of the standard phylogenetic inference tools with all the tricks they use to accelerate and approximate likelihood calculations, to assess its true potential.

While I clearly see potential, and am willing to test this myself, I would just ask the authors to formulate the title and conclusions a little bit more carefully, for instance, the title could read:

"Harnessing machine learning shows potential to boost heuristic"

One reason why I am suggesting this is that the authors consider all SPR moves that can be applied to a given tree, while most current search algorithms only evaluate the likelihood of a very small fraction of the SPR moves and also typically do not conduct a full evaluation (optimizing all model params and especially the branch lengths of the entire tree) of the SPR trees. To this end, the numbers given on page 8 lines 180-181 might be misleading.

Another issue that requires implementation and testing in one of the standard programs is the likelihood over inference time convergence behavior which is asymptotic when the search is in regions of the search space with a high likelihood. So one open question here is how and if this asymptotic convergence behavior will be altered by integrating the machine learning predictions. I do not have any gut feeling about this, but it might be that this asymptotic phase that consumes the vast majority of the run time might either become shorter or longer.

Finally, the only weak point of the study is the relatively small number of taxa (up to 70) contained in the test and training datasets. I believe that at least adding one or two datasets with up to 1000 or 2000 taxa to show that this really also scales with the number of taxa would be very important.

I do understand that the number of trees increases dramatically here, so some form of subsampling, e.g., by considering just say 5% or 10% of the subtrees that can be pruned and only re-inserting them up to a certain radius of say 10 nodes away from the pruning branch (instead of considering all regrafting positions) would be reasonable.

Thus, I only have two major concerns about the paper:

1. The conclusions and title should be formulated a little bit more carefully
2. The authors need to at least provide some anecdotal evidence that their method scales to large numbers of taxa.

Therefore, I suggest acceptance, pending minor revisions.

Reviewer #2:

Remarks to the Author:

The paper presents a random forest (RF) model that, given 20 features, predicts the log-likelihood of a particular SPR move, represented by those features. By enabling accurate prediction of the likelihood of the resulting tree, the method promises to reduce the time needed for tree search. The paper is based on a nice premise and has tantalizing results. I think this line of work is both creative and potentially valuable. Unfortunately, for reasons that I describe below, the paper has not fully achieved its stated goals.

My main disappointment in reading the paper was finding out that the authors have not really built a tool that would perform the phylogenetic tree search using the method described. Instead, they have given us a teaser. The results suggest that if the RF model is incorporated in a phylogenetic tree search tool, we could get substantial running time improvements. But it is one thing to give enough data to suggest, and it's quite another thing to demonstrate advances. To convince the reader of the value of this approach, authors need three things. 1) Build a software tool using the model for performing the phylogenetic search. 2) Show in extensive simulation and empirical analyses that the method is much faster than the many competitors available (RAxML, IQTree, FastTree, VeryFastTree, etc.). 3) Show the tool is as accurate (or effective in finding the optimal point) as the traditional tools. Until I see those analyses, I would classify the work as premature, especially for a journal like Nature Communications.

Beyond the fact that the work appears premature, I have several specific major comments.

1. I applaud the authors for distinguishing their validation dataset from the training set. However, it was not clear to me what was the size and properties of the validation test case. How many species were in these alignments? What was the diameter? Are validation datasets just like the training set in terms of dataset properties, or are they sufficiently different? In particular, see the next comment.

2. I was disappointed to see the dataset size was between 7-70 species. This range is not the dataset size for which we worry about running time. We routinely infer trees with thousands to tens of thousands of species these days using standard methods. If the new method is to change the state-of-the-art, it should be *tested* on such size. It's fine if the training set is small, as long as the model performs well on validation data that are not small.

3. Presumably, if this model was to be implemented in a software tool, we would have a fixed RF that works for all types of datasets. The results did not convince me that the model would continue to have high accuracy across different dataset types. The authors did take some steps (correlation with some properties among the data they do have). However, the authors need to tell us more about the range of properties of the datasets used in training, testing, and validation. What are their diameters (or levels of sequence divergence)? How about happiness and deviations from ultrametricity? In particular, for the validation set, I want to make sure the breadth of phylogenetic data is represented.

3.a. Related to this point, I was hoping authors didn't filter out any of the data, but they did filter out data with very high gappiness (line 280). Would the model work well if tested on gappy alignments?

4. I had a bit of trouble with the measurement metrics used. The vast majority of SPR moves reduce the likelihood, and a handful increases them. For those that decrease the likelihood, the ranking is not consequential. So I wouldn't focus on the full rank correlation (Fig 2a). We mostly care about identifying good moves. Showing that the predicted best SPR is among the top 20% is somewhat convincing; however, something like 5% of likelihood changes is positive, so 20% is not that high. The true top ranking SPR is ranked in the top 20% often, which is good, but the number can be lower in a substantial percentage of times (note long tail in Figure 2b). Another metric that I really would like to

see is the accuracy of the predicted *sign* of the likelihood change. For this, authors can show both false positive and false negative. Lines 174-176 do this a bit for one dataset, but results are not very strong. The model seems to have missed $\sim(2248*8\%=)180$ SPR moves with positive signs, which is three times as many of the ($\sim97*66\%=) 64$ positives that it did find correctly. This is not such great results at first glance. Perhaps this is good enough for the purpose of a fast search. But unfortunately, the paper doesn't demonstrate that by implementing an actual search.

Minor:

Table 1, item 7, I couldn't understand the relevance of c_1 . Also, I didn't understand the approximate length.

Figure 3 seems unnecessary and not informative.

Much of the introduction can be removed or shortened. It focuses too much on things that any reader familiar with phylogenetics would know.

Reviewer #3:

Remarks to the Author:

Azouri et al. present a machine learning approach to predict the improvement in likelihood for phylogenetic trees after an SPR move has been applied to a given phylogenetic tree. The results presented suggest that the machine learning approach could boost tree searches significantly. This can be a very important contribution to the field of computational phylogenetics because tree searches are one of the most fundamental aspects.

Overall I find the manuscript well written and useful for the community. For myself, as a methods developer, I'm primarily interested in the contribution of the features and how this new knowledge can be used to improve our existing algorithms. Thus, you will see that my main comments focus on the features. For the majority of readers, the important message here will be that using machine learning can make phylogenetic inference faster (although this is not yet implemented in commonly used software). I think that such improvements could also be done without machine learning by using several features instead of a single feature in previous approach, but this simply goes into the debate when we should call the machine learning or not.

In general, the conclusion are justified by the presented results and the work is clearly novel. I'm not surprised by these results because there has been done too little work previously and the approach taken here makes a lot of sense. The statistical analysis are also appropriate for this work. I've detailed my major concerns below, which mostly focuses on the description of the method/features and my surprise that not a single feature could be left out to produce similarly good results. This is still surprising to me and there are unfortunately not enough results presented to convince me.

General comments:

The explanation of the chosen features was a bit too brief. Specifically regarding the fifth feature, how is it that resulting trees have a different sum of branch lengths? Did you also change the branch lengths? Aren't SPR moves only splitting and merging branches, but not actually changing their length? Next, I don't understand the feature "The sum of branches in the starting tree". How can this feature distinguish between any predicted tree? The same should be true for feature 2 "Longest branch". Isn't the longest branch the same for all trees because you used the same starting tree?

Given that I don't understand the feature about the sum of branch in the starting tree, I'm particularly wondering why leaving out this feature decrease the accuracy of your predictions (same for feature 2). Do you have any explanation?

Following up on this, is there any data to see for the results of the stepwise elimination procedure (paragraph starting from line 322). I'm very surprised that you were not able to remove a single feature. Can you explain this?

Could you provide a table/figure with the correlation values p when you leave out 1 feature, 2-features, ... and which feature was left out?

Regarding the paragraph starting on line 182: What would happen if you always followed your predicted trees? Does this mean that if you would follow the best predicted trees, that you would end up in XX% with the true ML tree? When would there be a need to actually compute the likelihood?

In line 229ff you claim that your predictions could be useful for more complex models. This is a very interesting and useful prospect. However, it would be very important to know how if the learned regression model for your features also works for more complex models. That is, if these features are trained using a more simple model, then are the predictions still accurate? The simplest way of exploring this could be to learn the features under a JC model and the compare the predictions with likelihoods under a GTR+Gamma+I model. I would love to see some even more complex test where the substitution model would change among branches, but this might be out of scope. At the minimum, you should either backup your claim or revise it so that it shows truthfully that this is unknown.

In the conclusion, do you suggest for common software packages to have a fixed set of learned features or should these features be trained for every single analysis? How should software packages be changed to use your machine learning approach when the software is used by a standard user?

Specific comments:

- in the barrack, please be more concise that you are talking about ML approaches.
- line 21: please be more specific what you mean by "various heuristic approaches".
- line 21: Do you have examples for "Such approaches ..."
- line 47: I disagree here that "usually inferred by maximizing the likelihood" because Bayesian inference is similarly popular. If you say "often", that would be fine with me.
- line 58: Is NJ truly "often inaccurate"? Is there a reference for this?
- line180: Did the likelihood computation took 40 seconds per neighbor or for all neighbors together? This is not really clear. I would be very surprised if it was per neighbor.
- line 212: I think that the "average p for each of the remaining features" is actually much lower. In Table S2, the best remaining feature has a $p=0.463$ and the next best feature a $p=0.258$. There are 17 more features, so the mean must be much lower.
- line 241: Some typo in "In addition,S in"
- line 255: Examples for Bayesian applications are the eSPR from Lakner et al 2008 and pruned FNPR from Höhna and Drummond 2012. Perhaps you could discuss a bit how your approach would extend this previous work.
- Legend of Figure S2: It says "three data attributes" but I only see two.

Sebastian Höhna

Reviewer #1

The authors present a machine learning framework to predict the log likelihood improvement of SPR moves in phylogenetic tree searches with the goal to avoid compute-intensive likelihood calculations by identifying promising candidate SPR moves. Overall, the paper is very well written and technically sound.

Having worked on phylogenetic tree search algorithms for over 15 years, I have little to complain about, since the vast majority of questions I had in mind or omissions I thought were made, were actually answered in the following paragraph.

We thank the reviewer for this positive feedback.

This approach might have a good potential for accelerating likelihood-based tree searches, but needs to be implemented and tested in one or several of the standard phylogenetic inference tools with all the tricks they use to accelerate and approximate likelihood calculations, to assess its true potential. While I clearly see potential, and am willing to test this myself, I would just ask the authors to formulate the title and conclusions a little bit more carefully, for instance, the title could read: "Harnessing machine learning shows potential to boost heuristic"

Following the reviewer's suggestion, we modified the title of the work to better reflect its content to: "Harnessing machine learning to guide phylogenetic-tree search algorithms"

One reason why I am suggesting this is that the authors consider all SPR moves that can be applied to a given tree, while most current search algorithms only evaluate the likelihood of a very small fraction of the SPR moves and also typically do not conduct a full evaluation (optimizing all model params and especially the branch lengths of the entire tree) of the SPR trees. To this end, the numbers given on page 8 lines 180-181 might be misleading.

Indeed, most current search algorithms shorten the search time by evaluating only a fraction of the possible moves and/or by re-evaluating only the mainly affected parameters in each move. While comparing the running time of the machine-learning approach to a full log-likelihood optimization of all neighboring trees does not represent its advantage over current state-of-the-art software, it corroborates the strength of this approach: the entire set of neighboring trees could be rapidly considered without having to take (potentially harmful) filtering measures. Notably, there are numerous heuristic approaches that can be considered to speed up the computation even more. Here, we exemplify the utility of our method on an example dataset, in which we evaluate the log-likelihood of the top 5% predicted neighboring trees, and we make an SPR move to the neighbor with the highest log-likelihood. We repeat this process until convergence is obtained (i.e., the current tree has higher log-likelihood than all its 5% predicted neighbors). Clearly other heuristics are possible, and these are included in the Discussion.

We note that the estimated running times of the machine-learning approach include branch-length optimization. Thus, we compare the running times of evaluating the log-likelihood of a single neighboring tree (with branch length optimization, but fixing the model parameters) against the running time of estimating the log-likelihood using machine-learning. Following the reviewer's comment, we added a clarification of this point both to the Results and the Discussion sections.

In the results (page 8) we now write:

"In addition, we measured the running time for evaluating the 2,462 neighboring trees for this example. The computation of the features and the application of the trained model for each neighbor took 2×10^{-4} seconds on average. The likelihood computation (with branch lengths optimization) took 0.15 seconds on average for each neighbor, roughly 750 times longer compared to the machine-learning algorithm."

Another issue that requires implementation and testing in one of the standard programs is the likelihood over inference time convergence behavior which is asymptotic when the search is in regions of the search space with a high likelihood. So one open question here is how and if this asymptotic convergence behavior will be altered by integrating the machine learning predictions. I do not have any gut feeling about this, but it might be that this asymptotic phase that consumes the vast majority of the run time might either become shorter or longer.

We agree that it is an open question whether the asymptotic convergence behavior will be altered by integrating the machine-learning predictions. Nonetheless, it is implied in our analyses that our trained model performs well also in regions of the search space with high likelihood. Following the reviewer's comment, in the revised manuscript we examine this point more closely by testing the performance of the machine-learning approach on all trees that are one step away from the maximum-likelihood tree. Indeed, for our example dataset (with 28 species), we generated all trees that are one step away from the maximum-likelihood tree. For each such a neighboring tree, we applied our machine-learning algorithm and tested whether the maximum-likelihood tree is included in the top five predictions (i.e., whether the trained algorithm places the maximum-likelihood tree among its top predicted moves). We show that for 98% of these one-step-trees, this was the case. These results suggested a good performance of the machine-learning algorithm in regions of the tree space with high likelihood, although we agree that a more systematic investigation is needed. These findings are detailed in the amended Results (page 9):

"Finally, we examined the algorithm performance when the initial tree is one step away from the maximum-likelihood tree. To this end, we applied the machine-learning algorithm for each of the 2,492 SPR neighbors of the maximum-likelihood tree. The model predicted the maximum-likelihood tree to be among the top five predictions in 98% of the cases."

In addition, in the Discussion (page 12) we now explicitly write that the convergence behavior is an open question:

"Further developments of the proposed methodology towards a complete search are possible. For example, we have not put effort in assessing the branch lengths associated with the inferred topology or in predicting log-likelihoods of trees under different parameters-optimization schemes. It is also interesting to further study how our approach generalizes to additional substitution models of evolution, such as amino-acid models codon models, and additional partition models^{33,34}. Furthermore, the convergence behavior in regions of the tree space with high likelihood requires more robust investigation than the anecdotal evidence we provided in this study. In addition, our algorithm was implemented using SPR moves only. The benefit of using additional types of tree rearrangement moves, such as nearest neighbor interchange (NNI)^{35,36} and tree bisection and regrafting (TBR)³⁷ should be evaluated."

Finally, the only weak point of the study is the relatively small number of taxa (up to 70) contained in the test and training datasets. I believe that at least adding one or two datasets with up to 1000 or 2000 taxa to show that this really also scales with the number of taxa would be very important.

I do understand that the number of trees increases dramatically here, so some form of subsampling, e.g., by considering just say 5% or 10% of the subtrees that can be pruned and only re-inserting them up to a certain radius of say 10 nodes away from the pruning branch (instead of considering all regrafting positions) would be reasonable.

Following this important comment, we added an alignment with a much larger number of species (403 taxa) to the validation set and generated all its 624,508 SPR neighbors. The results obtained on this dataset were added to the revised manuscript. On page 10 we write:

"We next evaluated the performance of the machine learning approach on a dataset with a much larger number of species (403) than those used for training (spanning 7-70 species). For this dataset, the starting neighbor joining tree has 624,508 SPR neighbors. In this case, the best move was among the seven top predictions, which falls within the top 0.001% predictions, and the best prediction was among the top 20 moves (within the top 0.003% possible SPR neighboring-trees); the overall correlation between the predicted and true rankings was 0.69."

Noteworthy, considering 5% or 10% of a larger dataset by filtering according to a certain radius of nodes, could impair the strength of some of the features our model is based upon. For example, if we only sample SPR moves in which the subtree that was cut is pasted only one or two nodes from the cutting point, the feature that quantified the distance between the pruned and regrafted location becomes almost non-informative. We additionally note that our computational resources did not allow us to compute in a reasonable timeframe the likelihoods of millions of trees containing thousands of species and thus we could not validate the applicability of our trained model on such a large number of trees.

Thus, I only have two major concerns about the paper:

- 1. The conclusions and title should be formulated a little bit more carefully**
- 2. The authors need to at least provide some anecdotal evidence that their method scales to large numbers of taxa.**

Therefore, I suggest acceptance, pending minor revisions.

We appreciate these suggestions, we accounted for these helpful remarks in our revision as detailed above.

Reviewer #2

The paper presents a random forest (RF) model that, given 20 features, predicts the log-likelihood of a particular SPR move, represented by those features. By enabling accurate prediction of the likelihood of the resulting tree, the method promises to reduce the time needed for tree search. The paper is based on a nice premise and has tantalizing results. I think this line of work is both creative and potentially valuable. Unfortunately, for reasons that I describe below, the paper has not fully achieved its stated goals.

My main disappointment in reading the paper was finding out that the authors have not really built a tool that would perform the phylogenetic tree search using the method described. Instead, they have given us a teaser. The results suggest that if the RF model is incorporated in a phylogenetic tree search tool, we could get substantial running time improvements. But it is one thing to give enough data to suggest, and it's quite another thing to demonstrate advances. To convince the reader of the value of this approach, authors need three things. 1) Build a software tool using the model for performing the phylogenetic search. 2) Show in extensive simulation and empirical analyses that the method is much faster than the many competitions available (RAxML, IQTree, FastTree, VeryFastTree, etc.). 3) Show the tool is as accurate (or effective in finding the optimal point) as the traditional tools. Until I see those analyses, I would classify the work as premature, especially for a journal like Nature Communications.

Beyond the fact that the work appears premature, I have several specific major comments.

We thank the reviewer for the many insightful suggestions. The aim of our manuscript is to present a proof-of-concept approach. Our study is the first to combine machine-learning algorithm within phylogenetic tree search for more than four taxa. In the revised version, we have extended the analyses, further demonstrating the potential of our approach. We have modified the manuscript title to better emphasize that we only show the potential to accelerate current tree search algorithms. We agree that integrating our approach within state-of-the-art existing software is important, and we aim to pursue this direction in the near future. Nevertheless, we believe that our study should motivate a crosstalk between the community of evolutionary biologists and machine-learning developers, and is thus a gateway for combining AI within various phylogenetic algorithms.

1. I applaud the authors for distinguishing their validation dataset from the training set. However, it was not clear to me what was the size and properties of the validation test case. How many species were in these alignments? What was the diameter? Are validation datasets just like the training set in terms of dataset properties, or are they sufficiently different? In particular, see the next comment.

Following the reviewer's comment, we now provide more details that reflect the characteristics of the analyzed datasets, such as: the number of species, total branch lengths, deviation from ultrametricity, MSA length, and percent gappiness (see Supplementary Fig. 1). Furthermore, in the revised manuscript we apply the trained model to make predictions over two additional datasets that substantially deviate from the training set: (1) a dataset of 403 species, which is much larger than the datasets used for training (7-70 taxa); (2) a partitioned alignment of 59 species, in which different regions of the MSA evolve under a different model (see below) and is much longer than the average training data (6,951 characters; 8 partitions). The high performance of the machine-learning algorithm on these two datasets demonstrates that accuracy can be obtained even for datasets that are very distinct from those used for training. Supplementary Fig. 1 of the revised manuscript contains the information regarding the attributes of the training and validation datasets:

"Supplementary Figure 1. The dependence between data attributes and Spearman correlation. The Pearson correlation between model accuracy, as measured by Spearman correlation coefficient (ρ), and several data attributes: (a) the six databases used (four for training and two for validation); (b) the number of taxa; (c) the level of divergence as measured by the total branch length; (d) the alignment length; (e) the percentages of gaps in the alignment, averaged across all sequences in the MSA; (f) the deviation from ultrametricity as measured by the MAD score, which quantifies departures from ultrametricity. Left panels represent the training data (4,200 samples), while right panels represent the validation data (1,000 samples). In (b-f) the ' r^2 ' represents the squared Pearson correlation between the variable in the x axis and the Spearman correlation score of our machine-learning algorithm, and the 'pval' represents the two-sided P value for the null hypothesis that the data are uncorrelated."

2. I was disappointed to see the dataset size was between 7-70 species. This range is not the dataset size for which we worry about running time. We routinely infer trees with thousands to tens of thousands of species these days using standard methods. If the new method is to change the state-of-the-art, it should be *tested* on such size. It's fine if the training set is small, as long as the model performs well on validation data that are not small.

Following this and other comments, we added an alignment with a much larger number of species (403 taxa) to the validation set and generated all its 624,508 SPR neighbors. We present the results obtained on this dataset separately. On page 10 we write:

"We next evaluated the performance of the machine learning approach on a dataset with a much larger number of species (403) than those used for training (spanning 7-70 species). For this dataset, the starting neighbor joining tree has 624,508 SPR neighbors. In this case, the best move was among the seven top predictions, which falls within the top 0.001% predictions, and the best prediction was among the top 20 moves (within the top 0.003% possible SPR neighboring-trees); the overall correlation between the predicted and true rankings was 0.69."

3. Presumably, if this model was to be implemented in a software tool, we would have a fixed RF that works for all types of datasets. The results did not convince me that the model would continue to have high accuracy across different dataset types. The authors did take some steps (correlation with some properties among the data they do have). However, the authors need to tell us more about the range of properties of the datasets used in training, testing, and validation. What are their diameters (or levels of sequence divergence)? How about happiness and deviations from ultrametricity? In particular, for the validation set, I want to make sure the breadth of phylogenetic data is represented.

As detailed in our reply to the first comment, we now provide the performance metrics, while binning the data according to different characteristics. We believe that integrating the algorithm within software tools is a worthy next step, and indeed, we initiated preliminary discussions with the authors of RaxML.

3.a. Related to this point, I was hoping authors didn't filter out any of the data, but they did filter out data with very high gappiness (line 280). Would the model work well if tested on gappy alignments?

We agree. In the revised version, we did not filter out alignments according to the gap frequencies, therefore, both the training and validation sets contain highly gappy alignments. The relevant lines were modified accordingly (page 13):

"We randomly selected datasets with 7 to 70 sequences and more than 50 sites, excluding alignments containing sequences that are entirely composed of gapped or missing characters."

4. I had a bit of trouble with the measurement metrics used. The vast majority of SPR moves reduce the likelihood, and a handful increases them. For those that decrease the likelihood, the ranking is not consequential. So I wouldn't focus on the full rank correlation (Fig 2a). We mostly care about identifying good moves. Showing that the predicted best SPR is among the top 20% is somewhat convincing; however, something like 5% of likelihood changes is positive, so 20% is not that high. The true top ranking SPR is ranked in the top 20% often, which is good, but the number can be lower in a substantial percentage of times (note long tail in Figure 2b).

We agree that it is most interesting to identify potential good moves. Following the modifications made in this revision, i.e., the increased variability of our training set, together with employing RaxML-NG instead of PhyML, the overall performance of the machine-learning prediction increased in comparison to the original manuscript. Specifically, the predicted best SPR move is now among the top 5% moves in 86% of the cases. It is among the top 10% in 95% of the cases (in the previous version of the manuscript, we reported this number to be 89%). Of note, in 99% of the cases it is among the best 25% of the trees. Thus, not only that the predicted best move is among the top-scoring tree neighbors, but the proposed machine-learning approach can also accurately detect the best move among its top-scoring predictions. The revised version of our manuscript (page 9) now contains another supportive example to this prospect:

"We next studied the applicability of the machine-learning algorithm within a straightforward tree-search heuristic. Starting from the neighbor joining tree, we evaluated the likelihood of all top predicted 5% SPR moves and then moved to the highest scoring tree. We repeated this process until no improvement in log-likelihood was obtained. The log-likelihood increased and the Robinson-Foulds (RF)¹⁶ distance monotonically decreased for 15 consecutive moves (Supplementary Fig. 5). This procedure probably recovered the global maximum-likelihood tree (the tree with the highest likelihood obtained when running PhyML, RaxML-NG, and our own implementation from multiple starting points)."

Another metric that I really would like to see is the accuracy of the predicted *sign* of the likelihood change. For this, authors can show both false positive and false negative. Lines 174-176 do this a bit for one dataset, but results are not very strong. The model seems to have missed $\sim(2248 \times 8\%)=180$ SPR moves with positive signs, which is three times as many of the $(\sim 97 \times 66\%)=64$ positives that it did find correctly. This is not such great results at first glance. Perhaps this is good enough for the purpose of a fast search. But unfortunately, the paper doesn't demonstrate that by implementing an actual search.

The machine-learning algorithm was optimized to find the set of features and the way they are combined so as to maximize the correlation between the predicted ranking and true ranking. The training optimization was not aimed to find the correct sign. If determining the sign was the main goal, one could train a machine-learning classifier instead of the regression model implemented in this study. Nevertheless, following the model revision, we achieved better performance on the example dataset in terms of avoiding "misclassification" of the moves with a positive sign (18 among the 19 moves that were predicted by the model to increase the likelihood, actually increased it). In the Results (page 8) we write:

"Moreover, according to our model, 19 and 2,443 SPR moves were predicted to increase and decrease the log-likelihood, respectively, and these predictions were true for 95% and 98% of these cases."

Minor:

Table 1, item 7, I couldn't understand the relevance of c_1 . Also, I didn't understand the approximate length.

Thank you for pointing out our textual mistake regarding the referenced subtree. It was supposed to refer to the subtree in Fig.1d (the resulting tree), and not to c_1 . In any case, we note that we slightly modified the set of features, and they are all described in more details in the supplementary material (page 13-14).

Figure 3 seems unnecessary and not informative.

Removed.

Much of the introduction can be removed or shortened. It focuses too much on things that any reader familiar with phylogenetics would know.

We made an effort to shorten the introduction while preserving the details needed for the more generally oriented readers.

Reviewer #3/ Sebastian Höhna

Azouri et al. present a machine learning approach to predict the improvement in likelihood for phylogenetic trees after an SPR move has been applied to a given phylogenetic tree. The results presented suggest that the machine learning approach could boost tree searches significantly. This can be a very important contribution to the field of computational phylogenetics because tree searches are one of the most fundamental aspects.

Overall I find the manuscript well written and useful for the community. For myself, as a methods developer, I'm primarily interested in the contribution of the features and how this new knowledge can be used to improve our existing algorithms. Thus, you will see that my main comments focus on the features. For the majority of readers, the important message here will be that using machine learning can make phylogenetic inference faster (although this is not yet implemented in commonly used software). I think that such improvements could also be done without machine learning by using several features instead of a single feature in previous approach, but this simply goes into the debate when we should call the machine learning or not.

In general, the conclusion are justified by the presented results and the work is clearly novel. I'm not surprised by these results because there has been done too little work previously and the approach taken here makes a lot of sense. The statistical analysis are also appropriate for this work. I've detailed my major concerns below, which mostly focuses on the description of the method/features and my surprise that not a single feature could be left out to produce similarly good results. This is still surprising to me and there are unfortunately not enough results presented to convince me.

Thank you for the positive and constructive feedback. Following the helpful comments, we included more detailed descriptions regarding the features we used. As detailed below, we accounted for every remark and revised the manuscript accordingly.

General comments:

The explanation of the chosen features was a bit too brief. Specifically regarding the fifth feature, how is it that resulting trees have a different sum of branch lengths? Did you also change the branch lengths? Aren't SPR moves only splitting and merging branches, but not actually changing their length?

Following the reviewer comments, we slightly modified the set of features and they are now better explained, both in the manuscript and in the Supplementary material. Specifically, this feature is not included in the model of the revised version. Indeed, an SPR move does not change the total branch lengths. We note that in the previous version of the manuscript we followed the way SPR moves are conducted in Hordijk & Gascuel (2005). In there, the total branch lengths of the resulting tree may differ from that of the starting tree.

Next, I don't understand the feature "The sum of branches in the starting tree". How can this feature distinguish between any predicted tree? The same should be true for feature 2 "Longest branch". Isn't the longest branch the same for all trees because you used the same starting tree? Given that I don't understand the feature about the sum of branch in the starting tree, I'm particularly wondering why leaving out this feature decrease the accuracy of your predictions (same for feature 2). Do you have any explanation?

Indeed, both features are the same for all neighboring trees derived from a specific starting tree. Hence, these features cannot distinguish among neighboring trees if used as single features. However, the training set is composed of a variety of different starting trees, and the other features that represent the neighboring-trees may depend on the features of the starting tree. For example, consider a single decision tree (one of many comprising the random forest). It may be the case that in one of the internal nodes of this decision tree there is a condition regarding the length of the longest branch. The subtree to the right of this node is likely to be different than the subtree to the left of this node, i.e., the contributions of the other features vary depending on the length of the longest branch. Indeed, if we remove this feature, the performance of the machine-learning algorithm decreases (see the feature selection analysis in the comments below).

Following up on this, is there any data to see for the results of the stepwise elimination procedure (paragraph starting from line 322). I'm very surprised that you were not able to remove a single feature. Can you explain this?

Following the reviewer's helpful comment, we now better describe the results of the stepwise elimination procedure. Indeed, the performance of the model decreases only moderately for the first 11 elimination iterations, and only then starts to drop more significantly. Specifically, there was no statistically significant difference in accuracy when using the best 15 features and the entire set of 19 features. We decided to proceed with the entire set of features, as the other four do not add significant running time to our feature extraction algorithm, because random forest handles noise very well and disregards the features that do not contribute, and because the performance with the entire set of features was the highest. We also added a new figure (Fig 3) that details our feature importance analysis. We also added this relevant explanation to the Methods (page 15):

"To examine whether the feature set could be reduced to enhance computational performance, we applied a backward stepwise elimination procedure⁵⁰. To this end, we began with the full set of 19 features. We then removed the feature with the minimal importance score and trained the random forest algorithm for the remaining features, to compute the ρ metric. We repeated this procedure, successively eliminating an additional feature with the minimal importance score (Fig. 3). The best ρ value was obtained when all the features were included. Only when using 14 or fewer features, a statistically significant reduction in accuracy was detected (P value < 0.02 and P value > 0.49, for one-way T-test for the means when using 14 and 15 features to 19, respectively). The results across the entire analyses are presented using the entire set of features."

Figure 3:

"Figure 3. Feature selection following a backward stepwise elimination procedure. The mean Spearman correlation coefficient obtained when using a decreasing number of features for our algorithm. The box shows the quartiles of the dataset while the whiskers extend to show the $1.5 \times$ IQR past the low and high quartiles. The table at the bottom elaborates the feature composition within each set of features, as determined by the backward stepwise elimination procedure. "

Could you provide a table/figure with the correlation values ρ when you leave out 1 feature, 2- features, ... and which feature was left out?

Added. See Fig. 3 for the correlation values for each set of features considered in the backward selection procedure, and for the order by which the features were eliminated.

Regarding the paragraph starting on line 182: What would happen if you always followed your predicted trees? Does this mean that if you would follow the best predicted trees, that you would end up in XX% with the true ML tree? When would there be a need to actually compute the likelihood?

There is a host of possible approaches to combine the machine-learning algorithm with existing heuristic searches. Following the reviewer's comment, we examined the following procedure: starting with the neighbor joining tree, evaluate the log-likelihood of the top 5% predicted neighbors based on the machine-learning algorithm (full branch lengths optimization). Move to the best neighbor among this set of neighbors and repeat this procedure if the log-likelihood is greater than the current log-likelihood, else stop. What we observed is that we could follow 15 moves without a drop in log-likelihood. In particular, the resulting tree after 15 moves was probably the global maximum-likelihood tree (the tree that received the highest log-likelihood among PhyML, RaxML-NG, and our own search). Additionally, it resulted in a monotonically decreasing Robinson-Foulds function, i.e., the topological distance to the maximum-likelihood tree improved gradually from 26 to 0. These results, presented now in Supplementary Fig. 5, imply what would happen if only evaluate the 5% highest ranking neighbors. Of note, this result was achieved on an example dataset and more analyses are needed to suggest general heuristic search guidelines. We report this new analysis in the Results (page 9):

"We next studied the applicability of the machine-learning algorithm within a straightforward tree-search heuristic. Starting from the neighbor joining tree, we evaluated the likelihood of all top predicted 5% SPR moves and then moved to the highest scoring tree. We repeated this process until no improvement in log-likelihood was obtained. The log-likelihood increased and the Robinson-Foulds (RF)¹⁶ distance monotonically decreased for 15 consecutive moves (Supplementary Fig. 5). This procedure probably recovered the global maximum-likelihood tree (the tree with the highest likelihood obtained when running PhyML, RaxML-NG, and our own implementation from multiple starting points)."

Supplementary Figure 5:

"Supplementary Figure 5. Example for an iterative chain of moves. Evaluation metrics for the convergence behavior of an iterative chain of trees towards the maximum-likelihood tree. The chain was initiated with the Neighbor-Joining tree reconstructed for the protein-coding algae dataset we used as an example throughout the manuscript (iteration #0). The line plots represent the Robinson-Foulds distance from the maximum-likelihood tree (in blue; left Y axis), and the log-likelihood of the tree obtained in each iteration (in orange; right Y axis)."

In line 229 you claim that your predictions could be useful for more complex models. This is a very interesting and useful prospect. However, it would be very important to know how if the learned regression model for your features also works for more complex models. That is, if these features are trained using a more simple model, then are the predictions still accurate? The simplest way of exploring this could be to learn the features under a JC model and the compare the predictions with likelihoods under a GTR+Gamma+I model. I would love to see some even more complex test where the substitution model would change among branches, but this might be out of scope. At the minimum, you should either backup your claim or revise it so that it shows truthfully that this is unknown.

To address this matter, we both revised the relevant statement in the Discussion and introduced additional analyses. We first followed the reviewer's suggestion and trained a machine-learning model under the simpler JC model. To that end, for each of the 4,200 empirical datasets we reconstructed NJ starting trees (based on JC distances) and optimized their branch lengths under the JC model. Next, we calculated the log-likelihood of all neighbors under the JC model (again following branch lengths optimization). We then trained a machine-learning model to predict the log-likelihood of these neighbors. Finally, we applied the new trained model to our validation data (consists of all neighbors of the 1,000 empirical datasets). However, likelihood computations of the validation data were based on the GTR+I+G model. This resulted in a very high correlation between the empirical likelihoods and the predicted ones (average $\rho = 0.88$; see the added Supplementary Fig. 4), emphasizing the robustness of the machine learning to model misspecifications. The analysis results are now detailed on page 7:

"We repeated the above learning and testing procedure with the Jukes and Cantor (JC) model, which assumes that all types of substitutions are equally likely and ignores among site rate variation. Thus, this model is substantially simpler than the GTR+I+G model used above. When both learning and testing were performed assuming the JC model, the accuracy of the machine-learning model was high (average $\rho = 0.89$), similar to the accuracy obtained for the GTR+I+G model (average $\rho = 0.91$). We also evaluated the performance when the training was performed under the JC model, and the test data comprised of log-likelihoods computed under the GTR+I+G model. Under these conditions, the accuracy was only slightly lower (average $\rho = 0.88$). Similar results were obtained when alternative accuracy metrics were considered (Supplementary Fig. 4). These results suggest that learning on an oversimplified substitution model, is not detrimental for discriminating among potential neighboring trees, even when the underlying model is more complex than that used for training."

This is also presented in the Discussion. On page 11 we now write:

"Furthermore, our research could grant the development of richer and more realistic substitution models, which are currently too computationally intensive to be considered within a tree-search procedure (e.g., a covarion model²⁵ for codon characters). This hypothesis is based on the partitioned dataset analyzed in our study, and on our experiment in which high performance was still observed when we applied a machine-learning model trained under the JC model²⁶ to data evaluated under GTR+I+G model."

And on page 12 we write:

"Further developments of the proposed methodology towards a complete search are possible. For example, we have not put effort in assessing the branch lengths associated with the inferred topology or in predicting log-likelihoods of trees under different parameters-optimization schemes. It is also interesting to further study how our approach generalizes to additional substitution models of evolution, such as amino-acid models codon models, and additional partition models^{33,34}."

Second, we directly exemplified the performance of our (original) model on a new, more complex, dataset, composed of a partitioned eight-genes alignment. In this analysis, we used a partitioned model, where each gene has its own set of estimated model parameters and the branch lengths of each gene are proportional to each other using a gene-specific scaling parameter. On this dataset, we applied the machine learning model, which was trained to analyze simpler, non-partitioned datasets. As presented in the Results, starting from the initial tree (reconstructed using neighbor joining), the best predicted neighbor was ranked 10th among the true ranking (0.0008 percentile), while the best possible move was ranked 3rd among the machine-learning predicted moves. This analysis is now presented on page 9:

"We further validated that the accuracy of our model remains high when applied to datasets that represent larger and more complex biological scenarios than the ones included within the data used to train and test our model. To this end, we analyzed a partitioned dataset, consisting of eight protein-coding genes belonging to 59 plant species, where each partition is characterized by a different set of GTR+I+G model parameters and the branch lengths of each partition are based on the proportional model¹⁷. In this case, the best move was among the top three predictions (i.e., within the top 0.025% predictions) and the best predicted move was the twelfth possible move (i.e., top 0.1% SPR moves); the overall correlation between the predicted and true rankings was 0.74."

In the conclusion, do you suggest for common software packages to have a fixed set of learned features or should these features be trained for every single analysis? How should software packages be changed to use your machine learning approach when the software is used by a standard user?

Training the machine-learning model requires that the log-likelihood scores are known for the training set. When the trained model is applied to unseen data, the ranking of the trees is predicted, thus running likelihood computations of all neighboring trees is redundant. Therefore, the model should be trained once on a large cohort of datasets and the trained model can then be used on any given data. Accordingly, when designing the proposed approach, we have made sure to train it on an extensive set of realistic scenarios. We have also tested its performance on new data that were obtained from a different source (the validation set) and thus we verified that it performs well on unseen data. In its current setting, incorporation of the proposed machine-learning model in current software packages could be done, for example, by first using it to predict the rank of neighboring trees and then applying likelihood computation to the top ranked neighbors only.

Specific comments:

- in the barrack, please be more concise that you are talking about ML approaches.

Fixed. This is now better explained in the abstract. We now write:

"Inferring a phylogenetic tree is a fundamental challenge in evolutionary studies. Current paradigms for phylogenetic tree reconstruction rely on performing costly likelihood optimizations. With the aim of making tree inference feasible for problems involving more than a handful of sequences, inference under the maximum-likelihood paradigm integrates heuristic approaches to evaluate only a subset of all potential trees. Consequently, existing methods suffer from the known tradeoff between accuracy and running time."

- line 21: please be more specific what you mean by “various heuristic approaches”.

Following this and the next comment, we rephrased the abstract. Now we write:

"With the aim of making tree inference feasible for problems involving more than a handful of sequences, inference under the maximum-likelihood paradigm integrates heuristic approaches to evaluate only a subset of all potential trees. Consequently, existing methods suffer from the known tradeoff between accuracy and running time."

- line 21: Do you have examples for “Such approaches ...”

We rephrased the abstract and hence this sentence was removed.

- line 47: I disagree here that “usually inferred by maximizing the likelihood” because Bayesian inference is similarly popular. If you say “often”, that would be fine with me.

Agreed, corrected.

- line 58: Is NJ truly “often inaccurate”? Is there a reference for this?

We now refer to a study that compared the accuracy of neighbor-joining method to that of parsimony, maximum-likelihood, and Bayesian approaches. Additionally, we rephrased this sentence as follows (page 3):

"The general approach for a maximum-likelihood heuristic search is to begin either with a random starting tree or with a starting tree obtained by rapid and generally less accurate methods such as Neighbor Joining^{4,5}."

- line180: Did the likelihood computation took 40 seconds per neighbor or for all neighbors together? This is not really clear. I would be very surprised if it was per neighbor.

Yes, it was per neighbor in the previous version when we used PhyML and optimized all model parameters. Now, when we use RAxML-NG instead and optimizing branch lengths (and not the substitution model parameters), the likelihood computation per neighbor is faster by a factor of ~750 (0.15 second on average). We updated the results accordingly. On page 8 we write:

"The likelihood computation (with branch lengths optimization) took 0.15 seconds on average for each neighbor, roughly 750 times longer compared to the machine-learning algorithm."

- line 212: I think that the “average p for each of the remaining features” is actually much lower. In Table S2, the best remaining feature has a $p=0.463$ and the next best feature a $p=0.258$. There are 17 more features, so the mean must be much lower.

When writing “average p ” we meant average p across the training set, i.e., an average p for each feature. This is now better explained in the text. We rephrased the referenced sentence accordingly and now we write (page 10):

"The average p across the training set for all the other features was below 0.285 (Supplementary Table 2)."

- line 241: Some typo in “In addition,S in”

Thanks, corrected.

- line 255: Examples for Bayesian applications are the eSPR from Lakner et al 2008 and pruned FNPR from Höhna and Drummond 2012. Perhaps you could discuss a bit how your approach would extend this previous work.

In the initial submitted version, we noted that the machine learning approach could be integrated into existing maximum-likelihood and Bayesian frameworks. This integration into ML tools is already feasible, for example by sampling the most promising trees. However, the integration of the machine learning approach into Bayesian Phylogenetics could be challenging and requires that the proposal distribution (i.e., the posterior ratio between the proposed and current tree) is estimated with sufficient accuracy. We believe our work provides the first step into such future implementation, but further research is surely needed here. We thus opted to discard this possible implementation within the Bayesian paradigm as more thorough research is needed in this aspect.

- Legend of Figure S2: It says “three data attributes” but I only see two.

Thank you for pointing this inaccuracy in the legend. This figure was updated, and in the legend we now write:

"Supplementary Figure 1. The dependence between data attributes and Spearman correlation. The Pearson correlation between model accuracy, as measured by Spearman correlation coefficient (ρ), and several data attributes: (a) the six databases used (four for training and two for validation); (b) the number of taxa; (c) the level of divergence as measured by the total branch length; (d) the alignment length; (e) the percentages of gaps in the alignment, averaged across all sequences in the MSA; (f) the deviation from ultrametricity as measured by the MAD score, which quantifies departures from ultrametricity. Left panels represent the training data (4,200 samples), while right panels represent the validation data (1,000 samples). In (b-f) the ' r^2 ' represents the squared Pearson correlation between the variable in the x axis and the Spearman correlation score of our machine-learning algorithm, and the 'pval' represents the two-sided P value for the null hypothesis that the data are uncorrelated."

Reviewers' Comments:

Reviewer #1:

Remarks to the Author:

The authors have provided convincing replies to the comments made by all three reviewers (which showed substantial overlap) and have adapted large parts of the manuscript accordingly. They have also added a large set of new experiments.

From my point of view, this paper should be published now.

Alexandros Stamatakis

Reviewer #2:

Remarks to the Author:

In the revised manuscript, the authors have improved their method, addressed some of the reviewer's concerns, and have left others for future work.

I was excited to see:

- The accuracy of the method has improved a lot since the last version. In particular, it now finds the best move much more often than previously (lines 137-140, and 203-206). The updated results now seem reasonably accurate whereas the old accuracy seemed low.
- There are several new analyses that start to address some concerns raised by reviewers.
- The tone is changed to make it more clear that results are suggestive and need more work.

These are all positive improvements. Beyond these, I have some remaining general critiques and some specific requests. The general critiques may simply be a matter of opinion so I do not insist on a change in response.

==== General critique

* My main critique, that the method is not available yet and hence all the results are suggestive at best, remains. The addition of Figure S5 helps because it demonstrates that at least some search strategy works on one dataset. As the authors understand, this is very preliminary data. But it is better than no indication.

* I still think the ranking correlation is not the best objective function to use, especially if computed over all data points. For their future work, I suggest authors think about a weighted ranking score (giving more weights to highly ranked moves) or another score that seeks to detect the best moves. Again, the ranking among low-rank SPR moves does not give us anything useful. I don't think authors can change that at this stage, but for future work, they may find this suggestion useful.

* Relatedly, the new analyses on large numbers of species and large numbers of genes show mediocre correlation coefficients (0.74 and 0.69). If we cared about fully ranking, these results would not be that good. On those datasets, the authors first highlight the percentage of highly ranked moves, which is high. Thus, I believe authors do recognize that top-ranking moves, not a fully ordering of all SPR moves, are what matters. I wish they focused on those metrics everywhere (not just the large datasets where correlation coefficients are not as great).

==== Specific requests

Line 152. Similar to test data, please report for validation data: (1) the rank of this best move within the predicted ranking; (2) the rank of the predicted best move within the true ranking, as obtained according to the full likelihood optimization. From Figure S2, I gather performance is appreciably lower than the testing set (though it is still decent).

- Also, instead of reporting numbers in the sentence, perhaps you can add a table that shows these two measures for several thresholds *both for validation and testing data* (just a suggestion).

Figure S2 is very important; perhaps more than Figure 2. Please add Figure S2 to Figure 2 as a new row (so three more panels). That will make it clear to readers that in the validation set, part (b) does experience a drop in accuracy when you go from testing to validation set. Right now, only the most careful of the readers will see that drop.

- Space permitting, you may be able to also add S4 to Figure 2 (just a suggestion).

I also think Figure S5 may be a nice addition to the main paper (just a suggestion).

Reviewer #1/ Alexandros Stamatakis

The authors have provided convincing replies to the comments made by all three reviewers (which showed substantial overlap) and have adapted large parts of the manuscript accordingly. They have also added a large set of new experiments.

From my point of view, this paper should be published now.

Thank you for this affirmative feedback.

Reviewer #2

In the revised manuscript, the authors have improved their method, addressed some of the reviewer's concerns, and have left others for future work.

I was excited to see:

- The accuracy of the method has improved a lot since the last version. In particular, it now finds the best move much more often than previously (lines 137-140, and 203-206). The updated results now seem reasonably accurate whereas the old accuracy seemed low.
- There are several new analyses that start to address some concerns raised by reviewers.
- The tone is changed to make it more clear that results are suggestive and need more work.

These are all positive improvements. Beyond these, I have some remaining general critiques and some specific requests. The general critiques may simply be a matter of opinion so I do not insist on a change in response.

We thank the reviewer for this positive feedback.

==== General critique

* My main critique, that the method is not available yet and hence all the results are suggestive at best, remains. The addition of Figure S5 helps because it demonstrates that at least some search strategy works on one dataset. As the authors understand, this is very preliminary data. But it is better than no indication.

* I still think the ranking correlation is not the best objective function to use, especially if computed over all data points. For their future work, I suggest authors think about a weighted ranking score (giving more weights to highly ranked moves) or another score that seeks to detect the best moves. Again, the ranking among low-rank SPR moves does not give us anything useful. I don't think authors can change that at this stage, but for future work, they may find this suggestion useful.

* Relatedly, the new analyses on large numbers of species and large numbers of genes show mediocre correlation coefficients (0.74 and 0.69). If we cared about fully ranking, these results would not be that good. On those datasets, the authors first highlight the percentage of highly ranked moves, which is high. Thus, I believe authors do recognize that top-ranking moves, not a fully ordering of all SPR moves, are what matters. I wish they focused on those metrics everywhere (not just the large datasets where correlation coefficients are not as great).

We agree that a scoring method such as a weighted ranking metric could strengthen the reported results. We have tried to simplify the interpretability of our outcomes throughout this proof-of-concept study by presenting the most elementary metrics, yet we appreciate the suggestions raised by the reviewer, and we will clearly consider these in our future work.

=====Specific requests

Line 152. Similar to test data, please report for validation data: (1) the rank of this best move within the predicted ranking; (2) the rank of the predicted best move within the true ranking, as obtained according to the full likelihood optimization. From Figure S2, I gather performance is appreciably lower than the testing set (though it is still decent).

Also, instead of reporting numbers in the sentence, perhaps you can add a table that shows these two measures for several thresholds *both for validation and testing data* (just a suggestion).

Done. We now detail these results in Supplementary Table 1:

Data considered for accuracy evaluation	Metric (i) with $t = 10\%$	Metric (i) with $t = 25\%$	Metric (ii) with $t = 10\%$	Metric (ii) with $t = 25\%$
Testing data	56%	88%	91%	98%
Validation data	44%	79%	90%	98%

Accordingly we added a reference to Table S1 on page 7:

"The results obtained when alternative accuracy metrics were considered are detailed in Supplementary Table 1 and Supplementary Fig. 3b-c. These results suggest that learning on an oversimplified substitution model, is not detrimental for discriminating among potential neighboring trees, even when the underlying model is more complex than that used for training."

Figure S2 is very important; perhaps more than Figure 2. Please add Figure S2 to Figure 2 as a new row (so three more panels). That will make it clear to readers that in the validation set, part (b) does experience a drop in accuracy when you go from testing to validation set. Right now, only the most careful of the readers will see that drop.

Done. Figure 2 now contains two panels (a and b). Each panel contains the three metrics we use (1-3).

- Space permitting, you may be able to also add S4 to Figure 2 (just a suggestion).

We followed the three suggestions above regarding the representation of this section's findings, yet we believe it is more organized to present this figure (previously Supplementary Fig. 4, now Supplementary Fig. 3) in the supplementary material, together with the details of this analysis.

I also think Figure S5 may be a nice addition to the main paper (just a suggestion).

Done. Figure S5 is Figure 3 in the main paper.